# Overcoming the Impact of Hypoxia in Driving Radiotherapy Resistance in Head and Neck Squamous Cell Carcinoma

**DOI:** 10.3390/cancers14174130

**Published:** 2022-08-26

**Authors:** Rhianna M. Hill, Sonia Rocha, Jason L. Parsons

**Affiliations:** 1Department of Molecular and Clinical Cancer Medicine, University of Liverpool, Liverpool L7 8TX, UK; 2Department of Molecular Physiology and Cell Signalling, University of Liverpool, Liverpool L69 7ZB, UK; 3Clatterbridge Cancer Centre NHS Foundation Trust, Clatterbridge Road, Bebington CH63 4JY, UK

**Keywords:** head and neck cancer, hypoxia, ionizing radiation, radiotherapy, radioresistance

## Abstract

**Simple Summary:**

Hypoxia (reduced oxygen availability) is common in the majority of tumours, including head and neck cancer, and it occurs due to an imbalance between oxygen supply and demand. One of the key problems with hypoxia in tumours is that these areas are more resistant to radiotherapy treatment, which in turn leads to a poor prognosis of patients. It is important that new therapeutic techniques in combination with radiotherapy are developed to overcome hypoxia within the tumour to increase patient survival. This review aims to consolidate our current understanding of how hypoxia leads to radioresistance in head and neck cancer, and discuss past and future strategies to overcome this.

**Abstract:**

Hypoxia is very common in most solid tumours and is a driving force for malignant progression as well as radiotherapy and chemotherapy resistance. Incidences of head and neck squamous cell carcinoma (HNSCC) have increased in the last decade and radiotherapy is a major therapeutic technique utilised in the treatment of the tumours. However, effectiveness of radiotherapy is hindered by resistance mechanisms and most notably by hypoxia, leading to poor patient prognosis of HNSCC patients. The phenomenon of hypoxia-induced radioresistance was identified nearly half a century ago, yet despite this, little progress has been made in overcoming the physical lack of oxygen. Therefore, a more detailed understanding of the molecular mechanisms of hypoxia and the underpinning radiobiological response of tumours to this phenotype is much needed. In this review, we will provide an up-to-date overview of how hypoxia alters molecular and cellular processes contributing to radioresistance, particularly in the context of HNSCC, and what strategies have and could be explored to overcome hypoxia-induced radioresistance.

## 1. Introduction

Hypoxia is defined as reduced oxygen availability, either due to reduced oxygenation and/or increased oxygen demand, and is common in most solid tumours, including HNSCC [1,2]. The high proliferation rates of the tumour cells increase the demand for oxygen from the blood supply. This demand cannot be matched by the vasculature, so the tumour stress response encourages neovascularisation. However, poor vasculature is a common feature in tumours due to the fast development of disorganised and leaky vessels that vary in shape and diameter [3]. As a result, the oxygen delivery rate is rarely met so most tumours contain hypoxic areas. Most likely, an oxygen gradient occurs from the periphery to the centre, with a reduction in oxygen tension as the distance from the perivascular areas increases. The oxygen concentration in HNSCC tumour tissues has been shown to average between 1.3–1.9%, compared to normal tissue having an average of 5.3–6.7% [4]. It could be postulated that hypoxia would be detrimental to tumours since cells rely on oxygen. However, since the prevalence of hypoxia is so high in tumours, strong selection pressures result in cellular adaptations. Furthermore, hypoxia increases radioresistance which, coupled with the cellular adaptations, leads to a more aggressive phenotype. In fact, the importance of oxygen in radiotherapy treatment has been long known. In 1953 Gray et al. revealed the negative correlation between hypoxia and radiotherapy success [5]. The study estimated that severely hypoxic tissues require a radiation dose three times greater than that of normoxic tissues to create a similar level of damage. Hypoxia is now well established as a negative prognostic factor in the radiotherapy response of solid tumours, such as HNSCC, and several different strategies have been sought to overcome this in an attempt to lead to more effective treatment.

This review will summarise our current understanding of the impact of hypoxia on the effectiveness of radiotherapy, with a particular focus on HNSCC, and the role of hypoxia-inducible factor (HIF) in mediating this response. We will also review past and current clinical and preclinical strategies that have been investigated to overcome hypoxia-induced radioresistance, including hypoxic radiosensitisers, targeting HIF and the cellular DNA damage response (DDR), and utilising high linear energy transfer (LET) radiation.

## 2. Head and Neck Squamous Cell Carcinoma

Head and neck cancers primarily begin in squamous cells which line the mucosal surfaces of the larynx, pharynx and oral cavity and collectively are known as HNSCC. The worldwide incidence of HNSCC accounts for ~800,000 new cases each year, making it the sixth most common cancer type [6]. The global burden of HNSCC is rising and cases are estimated to increase by ~34% by 2030 [7]. However, survival rates have increased in recent years, with a 5-year survival of ~66% being evident across all age groups. Despite these improvements, patients still suffer detrimental side effects from treatment, such as dysphagia and odynophagia, which significantly reduce the patient’s quality of life [8]. The side effect burden on survivors is staggering, with a high proportion (63.4 cases per 100,000) resulting in suicide as a consequence of their treatment [9].

The major risk factors for HNSCC include excessive alcohol consumption, smoking and human papillomavirus (HPV) infection [10,11]. Smoking increases the chance of driver mutations, such as in the transcription factor TP53, which is a key tumour suppressor highly mutated in HNSCC [12,13]. The role that alcohol plays in the development of HNSCC is less well-understood, however it has been linked with a variety of somatic copy number alterations which can contribute to tumorigenesis [14]. In terms of HPV, this accounts for ~23% of all HNSCC cases worldwide, and HPV type 16 is the main strain involved in the development of HPV-positive HNSCC, particularly of the oropharynx [15]. Interestingly, HPV-positive HNSCC have a better prognosis and disease-free survival than patients with HPV-negative disease due to an increased response to radiotherapy and chemotherapy [16,17]. This highlights significant differences in the clinicopathological and biological characteristics of the two HNSCC subtypes. The HPV viral genome encodes two different types of structural genes, most commonly categorised as early and late genes [18]. In relation to HNSCC, the early E6 and E7 genes are the most important as these encode oncoproteins. E6 promotes ubiquitylation-dependent degradation of TP53, consequently leading to dysregulation of the cell cycle checkpoint required for the cellular stress response [19]. Similarly, E7 causes uncontrolled cell cycle progression through the degradation of another tumour suppressor protein, retinoblastoma (Rb) [20]. Ultimately, the presence of these oncoproteins leads to uncontrolled proliferation and the induction of genomic instability. Treatment for HNSCC typically involves the use of radiotherapy (ionising radiation; IR), which can be coupled with surgery and/or chemotherapy. As IR passes through the body, it deposits energy which induces damage to biomolecules within the cell, particularly the principal target which is DNA [21].

## 3. Impact of Hypoxia

### 3.1. Hypoxia and Radiotherapy

Clinically, the adverse impact of hypoxic tumours on the radiotherapy effectiveness of HNSCC patients has been clearly observed. In the 1990s, it was reported that HNSCC patients with hypoxic tumours had a significantly reduced disease-free survival following radiotherapy, compared to patients with less hypoxic tumours [22,23]. This was further demonstrated in an international multi-centre study of HNSCC patients which linked that a low pre-treatment measure of tumour oxygen correlated with a poorer prognosis [24]. Additionally, HNSCC patients with a low pre-treatment oxygenation status had a reduction in loco-regional tumour control following radiotherapy compared to well oxygenated tumours [25]. Preclinical research has also demonstrated that HNSCC cells in culture exposed to hypoxic conditions are more radioresistant than their normoxic counterparts [26]. Evidently, tumour hypoxia is particularly problematic in HNSCC patients and the reliance on radiotherapy as one of the major treatment options for these patients highlights the importance of a continued focus on modulating and improving this response. Not least will this lead to an improvement in patient survival and quality of life, but also reduce the financial burden of unsuccessful treatments.

The purpose of radiotherapy, or IR, is to create lethal DNA damage to the tumour cells. DNA base damage, sites of DNA base lose (apurinic/apyrimidinic sites) and DNA single strand breaks (SSBs) are the predominant lesions induced by IR. DNA double strand breaks (DSBs) and complex DNA damage (CDD), containing two or more lesions generated within close proximity of the DNA (1 helical turn), are less frequently induced although these are the most lethal type of damage due to their difficult nature for repair [27]. IR can directly damage the DNA, or more commonly in the case of conventional X-ray (photon) radiotherapy cause indirect damage through the production of highly reactive free radicals [28]. Hypoxic conditions can impact on these processes by firstly reducing the amount of hydroxyl free radicals produced, resulting in less DNA damage. Secondly, in a normoxic environment the free radicals generated by IR react with oxygen to form peroxyl radicals, which induces DNA damage that is difficult for cells to repair [28]. The DNA damage is described as being ‘fixed’ and which is known as the oxygen fixation hypothesis (OFH) [29]. A lack of oxygen leads to a reduction in fixed DNA damage and leads to a greater ability for the damage to be repaired, hence the increased radioresistance. The heterogeneity of tumour hypoxia adds an additional challenge. The duration and degree of hypoxia is rarely static and induces different radiobiological effects on the tumour cells. For example, an increased radioresistance of three HNSCC cell lines (SKK, FaDu and UT SCC5) was demonstrated when the cells were exposed to hypoxia (0.1% oxygen) prior to irradiation [30]. However, exposing the cells to hypoxia for an extended period (1% oxygen for 4–5 days) prior to irradiation, revealed varying cell line-specific results on radioresistance [30]. This highlights how the concentration and duration of hypoxia can impact on the radiotherapy response in tumour treatment. Subsequently, this variation will continue to increase selection pressures on surviving cells and generate even more malignant and aggressive phenotypes within tumours. There is therefore a need to further understand the hypoxic cellular survival mechanisms which could play a role in radioresistance. 

### 3.2. Hypoxia Inducible Factor (HIF)

Back in 1992, a nuclear factor was identified that exhibited a 7-fold transcriptional increase in hypoxic conditions [31]. This transcription factor was later described as HIF and the Nobel prize in Physiology or Medicine was subsequently awarded in 2019 to William Kaelin, Sir Peter Ratcliffe and Gregg Semenza for their collective research [32]. The HIF proteins act as heterodimers and induce the transcription of numerous genes involved in facilitating cell survival under hypoxic conditions. The HIF complex consists of an α subunit (HIF-1α, HIF-2α and HIF-3α), and the β subunit (HIF-1β) which is also known as aryl hydrocarbon nuclear translocator (ARNT) [33,34,35]. The master transcription factor in the hypoxic response is the HIF-1α/HIF-1β complex, described as HIF-1 and is an endogenous marker of hypoxia.

HIF-1α is constantly expressed by cells, where its regulation occurs at the post-translational level (Figure 1). Under normoxic conditions, HIF-1α is targeted for degradation via the oxygen dependent degradation domain (ODDD) with a half-life of ~5 min [36]. Firstly, hydroxylation of prolyl sites in the ODDD occurs via prolyl-4-hydroxylases (PHDs), which are 2-oxoglutarate and ferrous iron-dependent dioxygenases that depend on oxygen for the hydroxylation of the proline residues [35]. The hydroxylation of HIF-1α at prolines 402 and 564 increases the affinity of the protein for the tumour suppressor protein von Hippel-Lindau (VHL) [37,38]. HIF-1α interacts with the VHL ubiquitylation complex, which induces its degradation via the ubiquitin-proteasome pathway [35,37]. However, under hypoxic conditions, this suppresses HIF-1α hydroxylation and subsequent degradation leading to its accumulation. A second enzyme, factor-inhibiting HIF (FIH), catalyses the oxygen dependent asparaginyl hydroxylation in the C-terminal transcriptional activation domain [39]. This hydroxylation blocks the interaction of HIF-1α with the p300/CBP transcriptional co-activator proteins [39]. Collectively therefore, there are two key HIF hydroxylation events, with PHDs regulating HIF levels and FIH controlling HIF activity. In vitro experiments, with HeLa cells, have shown that HIF-1α becomes stabilised after as little as 3 h of hypoxic exposure, and an exponential increase in expression of HIF-1α occurs when the oxygen concentration falls below 6% oxygen [33,40].

Unlike HIF-1α, HIF-1β is regarded as constitutively expressed within cells and does not contain an ODDD. It resides in the nucleus and serves as a binding partner to the HIF-α subunits. The HIF complex regulates target gene expression through binding to hypoxia response elements which aid hypoxic survival. Remarkably, greater than 2% of genes in the human genome are thought to be regulated by HIF-1α either directly or indirectly [41]. HIF upregulation is associated with pro-tumourigenic effects due to its primary role in regulating genes involved in apoptosis, cell division, metabolism and angiogenesis [42]. Consequently, HIF has been proposed to play a role in the therapeutic resistance and poor prognosis in patients with hypoxic tumour phenotypes.

### 3.3. Autophagy

Autophagy is an evolutionary preserved process, termed as ‘self-eating’. It has an important role in maintaining cellular homeostasis through removing damaged organelles and proteins, and also plays a key role in the cellular response to stress [43]. Although autophagy is generally considered a protective mechanism, it has more recently been identified as a cell death mechanism as excessive autophagic events can lead to cell death. Interestingly, and related to HNSCC, it has been shown in oral squamous cell carcinoma patients that expression of Beclin-1 (a key protein in autophagy) was associated with a lower survival rate and increased lymph node metastasis [44]. Moreover, autophagy has been linked with tumour survival under hypoxic conditions. For example, in three human cancer cell lines (MCF7, PC3 and LNCaP), it has been demonstrated that hypoxia-induced cell death was greater in cells deficient in performing autophagy [45]. However, further evidence is required to investigate this, particularly in HNSCC. There is also evidence that autophagy plays a role in the radioresistance of tumour cells, including those from breast, cervical, lung, and glioblastoma tumours, although the exact mechanism remains to be revealed [46,47,48,49]. Furthermore, evidence is emerging that autophagy may be a major contributor to hypoxic radioresistance through the HIF-1α dependent upregulation of various autophagic proteins including Beclin-1 and LC3-II [49,50,51]. There is, however, a lack of research examining the role of autophagy in hypoxic radioresistance specifically in HNSCC, which may in the future represent a pathway to exploit in overcoming treatment resistance.

## 4. The Cellular DNA Damage Response to IR

As previously mentioned, the critical cellular target of IR leading to its therapeutic effect is damage to the DNA. IR induces DNA damage directly or indirectly through the production of reactive oxygen species (ROS). A typical therapeutic dose of IR at 2 Gy is thought to generate ~3000 DNA lesions per cell [52]. On DNA damage induction, activation of the DNA damage response (DDR) and the consequent signalling cascade is co-ordinated by the ataxia-telangiectasia and Rad3 related (ATR) and ataxia-telangiectasia mutated (ATM) protein kinases. These kinases enable the cell to undergo DNA repair and cause activation of the checkpoint kinases 1 and 2 (CHK1 and CHK2), respectively, which stimulate cell cycle arrest [53]. ATR is mostly active during SSB repair, with ATM being a key transducer in response to DSBs. Interestingly, recent studies have revealed that the increased radiosensitivity of HPV-positive HNSCC is attributed to defects in the ability to repair DSBs [54,55,56].

### 4.1. Non-Homologous End Joining (NHEJ)

NHEJ is the dominant pathway utilised in DSB repair as it does not rely on a homologous template, therefore is active at all stages of the cell cycle [57]. Despite this advantage, it is more error-prone in comparison to homologous recombination (HR) as it restores the molecular integrity of the DNA, but does not always retain sequence integrity [58]. There are two major mechanisms of NHEJ, classical NHEJ and alternative NHEJ. During classical NHEJ, DSBs are first recognised through the phosphorylation of the histone variant H2AX (designated γH2AX) by either ATM or ATR. This then promotes binding of the Ku70/80 heterodimer to the ends of the DNA (Figure 2), followed by the recruitment of the DNA-dependent protein kinase catalytic subunit (DNA-PKcs). DNA-Pkcs undergoes autophosphorylation and forms a nuclease complex with Artemis, whilst the DNA polymerases, Pols μ and λ, synthesise new nucleotides and the DNA undergoes ligation via the ligase complex containing XLF, XRCC4 and DNA ligase IV [58]. Conversely during alternative NHEJ, poly(ADP-ribose) polymerase-1 (PARP-1) binds to the DSB ends (Figure 2), and induces poly(ADP ribosyl)ation which primes the DNA for resection via the MRE11-RAD50-NBS1 (MRN) complex and carboxy-terminal binding protein-interacting protein (CtIP) [59]. This generates areas of microhomology and then finally the DNA undergoes ligation via the XRCC1-DNA ligase IIIα complex [59]. 

### 4.2. Homologous Recombination (HR)

In contrast to NHEJ, the HR pathway is an error free repair mechanism. It is predominantly initiated during the S/G2 phase of the cell cycle as it requires homologous sister chromatids to be present. Briefly, DSBs are sensed by the MRN complex which induces DNA end resection in a BRCA1-dependent manner (Figure 2) [53,60]. This generates 3′-DNA single stranded regions which are stabilised by replication protein A (RPA) [61]. BRCA2 then recruits RAD51 to the single stranded DNA which displaces RPA. The role of RAD51 is to match the sequence of the broken DNA strand to a homologous sequence within the DNA helix [62]. Holliday junctions are then generated following strand invasion and DNA synthesis, which are acted on by resolvases to complete DSB repair [63].

Both NHEJ and HR are vital mechanisms for cell survival and maintaining genetic integrity, and defects in key proteins within these pathways can lead to an accumulation of genetic mutations ultimately contributing to cancer development. Conversely, tumour cells with efficient DDR pathways are capable of efficiently repairing damage induced by IR and can develop radioresistance. Due to the high reliance on radiotherapy for the treatment of HNSCC, radioresistance is a major obstacle as it leads to poor prognosis and an increased treatment burden for patients [64], and one of the major contributors to this phenotype is the presence of hypoxia.

## 5. Overcoming Hypoxic Radioresistance in HNSCC

Overcoming hypoxic radioresistance in HNSCC patients is essential in optimising the effectiveness of radiotherapy treatment, and there is an ongoing need for novel strategies to be developed. Unsurprisingly, there have been many attempts over the last five decades to modulate hypoxic conditions in clinical trials (summarised in Table 1). Other strategies have and are currently being investigated (summarised in Table 2), particularly in preclinical experiments, and which will also be summarised below.

### 5.1. Hyperbaric Oxygen and Carbogen

Hyperbaric oxygen was one of the first techniques to be examined in combination with radiotherapy to treat HNSCC patients back in the 1970s. Patients were exposed to hyperbaric oxygen prior to radiotherapy with the aim of improving tumour oxygenation. Initial results appeared promising, with increased survival, lower recurrence rates and some evidence of tumour control in HNSCC patients [108]. Conversely, there was an increase in severe radiation tissue injury and seizures which has hindered its progression into routine clinical practice [108]. Despite the limitations of this method, it provides important knowledge that modulating tumour hypoxia can be beneficial to HNSCC patients.

Similarly, carbogen has been studied in conjunction with radiotherapy to overcome hypoxic radioresistance. Carbogen comprises of 95% oxygen and 5% carbon dioxide. The Radiation Therapy Oncology Group in the 1970s enrolled 254 HNSCC patients onto a study but disappointingly, there was no evidence that administering carbogen prior to radiotherapy improved local or regional control of the disease [72]. However, the use of carbogen in conjunction with nicotinamide, prior to accelerated fractionated radiotherapy, has been shown to radiosensitise rodent tumours [109]. The rationale being that nicotinamides can prevent the intermittent closure of blood vessels and subsequently reduce tumour hypoxia [110]. A clinical trial utilised ARCON (accelerated radiotherapy with carbogen breathing and nicotinamide) in 215 advanced HNSCC patients, which was found to successfully improve both local and regional tumour control [73]. However, the initial dose of nicotinamide generated profound side effects in most patients and had to be reduced to a more tolerable level. A phase III clinical trial later investigated the use of ARCON in advanced laryngeal cancer patients, and found improvements in regional control but no difference in local tumour control [74]. Importantly it was observed that only patients whose tumours were hypoxic benefitted from ARCON, with no improvement found in well oxygenated tumours. This demonstrates the importance of targeting hypoxic modifications to patients with confirmed hypoxic tumours, rather than adopting a more universal regime. Personalisation of treatments is key and should be the driving force for future therapeutic developments.

### 5.2. Nitroimidazoles

Nitroimidazoles have a long history of use as potential hypoxic radiosensitisers. Interestingly, many of these compounds are already licensed for use as antibiotics and antifungals. Nitroimidazoles are described as being oxygen mimetics when combined with radiotherapy, where they act to fix radiation induced DNA damage that is harder to repair subsequently leading to cell death [111]. One of the first nitroimidazole compounds to be investigated in HNSCC was misonidazole. Unfortunately, multiple trials did not show any significant differences between patients treated with misonidazole prior to radiotherapy, compared to the control group [75]. Furthermore, severe side effects were reported, which included peripheral neuropathy [75]. A similar compound, etanidazole, has similarly been found not to improve radiotherapy success in HNSCC patients and generated severe side effects [76].

Following the failure of misonidazole and etanidazole, a less toxic alternative called nimorazole was trialled. The Danish Head and Neck Cancer Study Group found that combining nimorazole with radiotherapy in HNSCC patients significantly improved local regional control and increased overall survival [77]. Unlike the previous compounds, the side effects were mild, with nausea and vomiting being the most common complication. Despite the success of this trial, the use of nimorazole is only routinely used in Denmark as a therapeutic option for HNSCC. It was later shown that only patients who exhibited expression of a larger subset of hypoxia genes benefited from nimorazole administration [112]. Again, this highlights the importance of assessing the hypoxic status of patient tumours, prior to administering hypoxic radiosensitisers. Due to its positive results and lack of side effects, nimorazole is currently being investigated in a phase III clinical trial in the UK, NIMRAD, which will further investigate the use of nimorazole as a radiosensitiser in HNSCC patients [78] and its results are highly anticipated.

### 5.3. Tirapazamine

Another drug which has gained attention in the fight against hypoxic radioresistance is tirapazamine (TPZ). This drug belongs to a class of bioreductive cytotoxic drugs and reportedly only targets hypoxic cells [113], allowing for their selective radiosensitisation and thus increasing normal tissue sparing. A phase II clinical trial of tirapazamine with radiotherapy in advanced HNSCC patients showed promising results [82]. The treatment was well tolerated by patients and showed encouraging tumour control. However, more recently, a phase III clinical trial in HNSCC patients demonstrated surprisingly that combining the chemotherapy drug cisplatin, TPZ and radiotherapy was no more successful than cisplatin and radiotherapy alone [83]. Importantly though, this study did not assess tumour hypoxic status prior to treatment. Therefore, a potential reason for its lack of success could be that the use of TPZ is redundant in patients whose tumours do not exhibit significantly hypoxic environments. Furthermore, as previously mentioned, there is a need to differentiate between the hypoxic status of tumours to gain a greater insight into the potential benefits of hypoxic radiosensitisers, such as TPZ.

### 5.4. Targeting the DDR

Hypoxia has been shown to contribute to radioresistance through modulating the efficiency of DDR mechanisms. Activation of key DDR transducers, ATM, ATR and DNA-PKcs, has been shown to occur in response to severe, radiobiologically hypoxic conditions (<0.1% oxygen) where significantly reduced radiosensitivity is observed, in addition to accumulation of γH2AX foci [114,115,116,117]. Initially, it was unclear what led to this initiation of the DDR in hypoxia given that it was shown to occur independently of DNA damage, and noticeably that the γH2AX foci were spread across the whole nucleus rather than as discreet sites commonly seen post-IR. However, accumulating data suggest that DDR activation is likely due to replicative stress. It was found that ATR activation occurs under radiobiological hypoxia in response to replication stress and not due to detectable DNA damage [116], and that the activation of ATR helps to maintain replication fork integrity and ultimately induces p53 dependent apoptosis [118]. This mechanism has been shown to be essential in maintaining cell proliferation in hypoxic conditions [119]. As a result, strong selection pressure for p53 mutations occurs within hypoxic tumour cell populations, suggesting overall a possible explanation for the increased radioresistance and aggressive phenotype observed in hypoxic tumours.

Despite the hypoxic activation of key DDR transducers, interestingly the downstream effectors do not appear to follow this trend. It has been suggested that HR mediated DNA repair is downregulated in hypoxic conditions [120]. For example, in a variety of cancer cell lines (MCF-7, A549, HeLa, SW480, A431, RKO and PC3) RAD51 protein levels were observed to be downregulated in radiobiologically hypoxic (<0.5% oxygen) conditions [121,122]. Furthermore, it was also confirmed in three HNSCC cell lines exposed to chronic hypoxia (1% oxygen) for 4–5 days prior to irradiation, that there was downregulation of RAD51 indicating a reduction in HR capability [30]. In terms of radiotherapy treatment, this initially appears advantageous given this reduction in DNA repair capability should lead to DNA damage persistence and cell death, although this conflicts with the fact that hypoxia is known to increase radioresistance. It has therefore been suggested that hypoxic cells predominantly rely on NHEJ, supported by an increase in activated DNA-PKcs [30]. The reliance on this more error prone DSB repair pathway likely leads to an accumulation of DNA damage and increased genetic instability, ultimately leading to cancer progression and therapy resistance due to selective advantage. Additionally, downregulation of HR by hypoxia (1% oxygen for 3–6 days) has been suggested to lead to activation of ATM and MRE11 [123], which could aid tumour radioresistance and survival. Importantly though, it should be noted that the many studies into the DDR and hypoxia have been conducted in severe hypoxic conditions (<0.1% oxygen). It is therefore difficult to compare responses where the length of hypoxic treatment and the percentage oxygen varies significantly. Distinctions between the impact on the DDR in mild and severe hypoxia are consequently lacking and are necessary to fully understand how this is affected across the whole tumour.

Due to the differences observed in the DDR in normoxia and hypoxia, DDR inhibitors could be a suitable target to overcome hypoxic radioresistance in HNSCC. DDR inhibitors in normoxia have shown to be effective radiosensitisers of HNSCC cells [53] but their use in hypoxic conditions has not been extensively studied. Nevertheless, some success has been reported, particularly in other tumour types. The reliance on ATR to protect replication fork integrity in radiobiological hypoxia is an attractive target. There have been some promising in vitro results showing that inhibition of ATR sensitised various tumour cell lines to severe hypoxia [84,85]. Furthermore, DNA-PKcs inhibition (KU57788 and IC87361) has shown success in radiosensitising hypoxic (0.2% oxygen) HNSCC cell lines [86] and additionally SN38023, which is a hypoxia activated prodrug which releases the DNA-PKcs inhibitor IC87361 preferentially in hypoxic cells, was demonstrated to selectively radiosensitise anoxic UT-SCC54 cells [87]. The latter method of selective radiosensitisation would be beneficial in overcoming hypoxic tumour radioresistance whilst decreasing normal tissue toxicity. PARP inhibitors have been shown to be synthetically lethal to tumour cells lacking HR capabilities, so this strategy could be effective in sensitising hypoxic tumour cells given the downregulation of HR under radiobiological hypoxia [88]. As an example, it has been demonstrated in prostate and non-small cell lung cancer cell lines that the PARP inhibitor ABT-888 radiosensitised hypoxic (0.2% oxygen) tumour cells, but which was similar to that observed in normoxic tumour cells [89]. However, caution should be taken as recently it has been found that mild hypoxia (2% oxygen) actually increased the resistance of HR-proficient and deficient cancer cells to PARP inhibition, whilst severe hypoxic conditions (<0.5% oxygen) increased sensitivity [90]. Nevertheless, it is clear that more substantial research into DDR inhibition of hypoxic cells is needed to examine the potential for successful tumour radiosensitisation.

### 5.5. Targeting HIF

Immunohistochemical studies have shown that HIF-1α overexpression is present in most cancers, and clinical studies have also revealed an inverse correlation between HIF-1α expression and overall patient survival [124]. In HNSCC, the expression of HIF-1α has been shown to vary dependent on tumour location. Tumour sites located at the base of the mouth appear to have stronger HIF-1α expression than tumours of the tongue [125]. A systematic review of HIF expression and HNSCC revealed that overexpression of HIFs was significantly associated with an increased mortality risk [91]. Studies have also linked HIF-1α to being a prognostic marker of radiotherapy response. Expression of HIF-1α in oropharyngeal cancer patients has been linked with a decrease in overall survival, primarily due to a reduced response to radiation treatment [92,126]. Furthermore, patient-derived primary oral squamous cell carcinoma cells, grown under normoxic conditions, revealed an increase in nuclear translocation of HIF-1α following exposure to γ-rays and a negative correlation between radiosensitivity and HIF-1α protein levels [127]. Furthermore, when these cells were subjected to HIF-1α siRNA, there was an observed increase in radiosensitivity.

In HNSCC xenograft models, it has been suggested that the restoration of radiosensitivity following HIF-1α knockdown, was due to targeting the HIF-dependent glycolysis pathway [128]. This pathway induces the HIF-1α-dependent switch to anaerobic metabolism, a crucial part of the hypoxic response. However, it has been argued that the inability of the cells to switch to anaerobic metabolism could impact tumour survival independent of IR, and therefore the decreased cell survival may not entirely be because of increased radiosensitivity. As an alternative model, HIF-1α has been suggested to induce radioresistance by increasing vascular endothelial growth factor (VEGF) and inhibiting p53 [129]. Furthermore, and following IR exposure, certain areas of the tumour were observed to become reoxygenated due to cell death and there were changes to the vasculature. Unexpectedly this reoxygenation caused a nuclear accumulation of HIF-1α in response to an increased production of ROS. This finding contradicted previously reported oxygen-dependent regulation of HIF [130], although it has been shown that ROS actually alters PHD function as well as increasing the transcriptional activation of HIF-1α [131]. Nevertheless, as a result, downstream HIF-1α target genes are likely to undergo a surge in translation following radiation. This could lead to an aggressive cancer phenotype and subsequent radioresistance.

Despite many clinical studies revealing that HIF-1α expression is linked with a poor prognosis in HNSCC, surprisingly some studies have revealed the opposite. In surgically treated HNSCC patients, HIF-1α was associated with improved disease free and overall survival [132]. As previously mentioned, high HIF-1α expression has been related to a poor outcome in oropharyngeal and laryngeal squamous cell carcinoma, however a better patient prognosis was shown with in patients with oral cavity tumours [133]. HPV status of HNSCC has additionally been shown to have an impact on HIF-1α expression. HPV-positive HNSCC cells were demonstrated to have greater HIF-1α protein levels in both normoxic and hypoxic conditions [134], and given that it is well known that patients with HPV positive HNSCC have a better prognosis compared to HPV-negative disease, this suggesting that HIF-1α may play a role in enhanced sensitivity to radiotherapy. Given these conflicting evidence, further evidence examining the link between HIF-1α, p53, relative hypoxia, tumour location and HPV status in the radiosensitivity of HNSCC is required.

The HIF pathway remains an attractive therapeutic target to improve tumour radiosensitivity, and therefore HIF inhibitors have been developed [93]. Focussing on HNSCC, very few studies have examined the combination of HIF inhibition with IR. Two studies by the same team have shown that the indirect HIF-1α inhibitors, BAY 84-7296 and BAY 87-2243, display some promising results where they act as small molecular inhibitors that block mitochondrial ROS formation, thus reducing HIF-1α activity [94,135]. In HNSCC xenograft models, it was demonstrated that administering each of these inhibitors prior to single dose or fractionated radiotherapy improved tumour control. Despite this, the use of HIF-1α inhibitors as radiosensitisers has been hindered by a lack of mechanistic knowledge. For example, in HeLa cells, utilising the HIF-1α inhibitor, YC-1, prior to IR did not improve radiosensitivity, but strikingly adding YC-1 after IR caused increased cellular radiosensitivity [136]. This highlights a gap in the knowledge of how and when HIF-1α may contribute to cellular radiosensitivity. In terms of clinical translation of HIF-1α inhibitors, this has been impeded by the lack of specificity of the inhibitors, as well as evidence suggesting that HIF-1α may actually be required to enhance radiosensitivity as discussed above [137]. Therefore, further preclinical research in HNSCC models is required to understand the radiobiological consequences of targeting HIF-1α.

Despite the major focus being on HIF-1α in relation to radioresistance, recent evidence has interestingly shown that HIF-2α may be just as important. There are clear similarities between both HIF-1α and HIF-2α isoforms which promotes the question of the purpose of both these being present, although evidence has shown non-redundant roles for the isoforms during embryonic development [138]. HIF-2α may also act as a compensatory subunit for a lack of HIF-1α, as HIF-2α expression appears to increase in HIF-1α knockout cells [139]. Nevertheless, a strong link between high HIF-2α expression and radioresistance has been shown in renal cell carcinomas in particular [140,141,142]. Furthermore, in HNSCC cell lines resistance to cetuximab and radiotherapy has been demonstrated to correlate with high HIF-2α expression and silencing of HIF-2α decreased clonogenic survival post-IR [143]. Tissue samples from HNSCC patients pre-IR treatment showed a significant association between high HIF-2α expression and poor local regional control with IR [144]. Despite this, the mechanism through which HIF-2α may influence radioresistance is not fully understood. Unlike HIF-1α, there has been progress in the development of a specific HIF-2 inhibitor, PT2385, for clinical use although only in renal cell carcinoma. Clinical trials using PT2385 as a monotherapy have shown promising results with no dose limiting toxicities reported [95], but its use in conjunction with radiotherapy and in other cancer types such as HNSCC still needs to be addressed. There needs to be a greater understanding of the interplay between HIF-1α and HIF-2α as well as examination of the expression levels of HIF-2 in HNSCC, as joint targeting may be more beneficial when attempting to overcome hypoxic radioresistance.

### 5.6. Immunotherapy

Radiotherapy not only causes cytotoxic effects on tumour cells, but it also exerts a radiation-induced immune response that can trigger radioresistance and enhance tumour survival [96]. In HNSCC patients subjected to chemo-radiotherapy, an upregulation in circulating inhibitory immune T_reg_ cells compared with untreated or surgically treated patients has been observed [145]. This coupled with hypoxic radioresistance generates a very radioresistant phenotype. There has been a lot of attention on the immune checkpoint receptor programmed death 1 (PD-1), which is involved in preventing autoimmunity and modulating T cell responses. PD-1 interacts with programmed death ligand 1 (PDL-1) to induce its inhibitory effects, and PDL-1 is upregulated in many tumours which prevents tumour detection by the immune system. Anti-PDL-1 antibodies have demonstrated high response rates and low side effects in advanced cancer patients [97], and combining anti-PDL-1 blockade with radiotherapy has been shown to induce primary tumour regression, as well as abscopal effects on distant tumours [96,146]. However, when nivolumab, a drug preventing the PD-1/PDL-1 interaction, was administered to HNSCC patients in conjunction with radiotherapy, there was no observed improvement in overall survival or response rate [147].

Interestingly, hypoxia has been shown to cause an upregulation of PDL-1 which is dependent on HIF-1α. Chromatin immunoprecipitation and luciferase reporter assays have shown direct binding of HIF-1α to a transcriptional hypoxia response element in the PDL-1 promoter [98], and hypoxia induced PDL-1 overexpression has been shown to be abrogated by a HIF-1α knockdown in glioma cells [99]. These data provide evidence for a relationship between PDL-1 and HIF-1α, and that hypoxia may play a role in altering tumour immunosurveillance thus enhancing therapy resistance. It also provides the rationale for dual targeting of both HIF-1α and PDL-1 to overcome hypoxic radioresistance, as well as preventing radiation induced immunosuppression. The use of immunotherapy has been shown to normalise tumour vasculature and increase perfusion, therefore reducing tumour hypoxia [100,148,149], and therefore the dual use of immunotherapy and radiotherapy could overcome hypoxic radioresistance.

With regard to HNSCC, there are currently over 40 clinical trials combining immunotherapy with radiotherapy [150]. Most of the studies so far have investigated the safety of the combination and presence of any adverse side effects, which have shown promising results. For example, a phase II clinical trial in HNSCC patients found that the anti-PD1 drug pembrolizumab combined with radiotherapy was well tolerated when compared to cetuximab and radiotherapy [151]. The main side effect reported was mucositis. The next question to be addressed is the efficacy of the combination treatment, results of which are highly anticipated, and which will assess the applicability of immunotherapy in HNSCC patients as an alternative therapeutic approach to overcome radioresistance.

### 5.7. High-LET and FLASH Radiotherapy

High-LET radiation is highly beneficial in overcoming hypoxic radioresistance as there is less reliance on indirect damage and a greater contribution to direct DNA damage induction which does not require oxygen. The utilisation of proton beam therapy (PBT) in cancer therapy is increasing, principally as this displays radiobiological advantages over photon radiotherapy. PBT is advantageous due to a lower entrance dose of radiation which spares normal tissues, and the dose can be targeted directly at the tumour site via the Bragg peak, thus reducing adverse treatment side effects [27]. However, there is an increase in ionisation density (LET) with PBT particularly at the distal end of the Bragg peak, which can lead to formation of CDD that is more difficult for cells. Indeed, using a 60 MeV proton beam it has been demonstrated in HNSCC cells that CDD is generated in relatively high-LET regions of the Bragg peak which persists for several hours post-irradiation [152,153]. Mathematical modelling based on a 62 MeV proton beam has predicted that PBT increases the relative biological effectiveness (RBE) under hypoxic conditions (2% and 0.1% oxygen), compared to conventional photons [154]. However, to our knowledge, there is no reported evidence of preclinical studies using HNSCC cell models that demonstrate that PBT can overcome hypoxic radioresistance, demonstrating a need for research in this area.

The use of carbon ion therapy has shown promising results in attempts to overcome hypoxic radioresistance. Carbon ion therapy is potentially more beneficial than conventional photons and PBT as it displays a significantly higher LET and therefore larger RBE, and a smaller oxygen enhancement ratio (OER) resulting in greater effectiveness against hypoxic tumours [101]. To date, there has been only a single clinical study in uterine cancer patients that has directly linked the efficacy of carbon ion therapy with tumour hypoxia. The results were promising, as they showed no differences in local regional control and disease-free survival between oxygenated and hypoxic tumours with carbon ion therapy [102]. Recent in vitro experiments have revealed encouraging results in non-small cell lung carcinoma. Carbon ion therapy effectively eradicated hypoxic (1% oxygen) tumour cells which was further enhanced when combined with DNA-PKcs inhibition [103]. However, the use of carbon ion therapy may be more cell specific than originally thought. Carbon ion therapy appeared to have different effects on the OER of two glioblastoma (GBM) cell lines under hypoxia (1% oxygen) [104]. This suggests that there may be tumour-dependent effects on the success of carbon ion therapy in overcoming hypoxic radioresistance. The authors suggested that the use of carbon ion therapy may be more effective when combined with additional therapeutics, such as targeting the HIF pathway to increase susceptibility of GBM cell lines even further [104]. Whilst initial results are promising results, further investigation into the use of carbon ions in overcoming the radioresistance of HNSCC specifically is required.

Another exciting avenue for the future of radiotherapy is the introduction of FLASH radiotherapy. FLASH delivers ultra-high dose rates (>40 Gy/s) in a much shorter time frame than conventional radiotherapy, and is beneficial because it has been shown to spare normal tissues from any damage without compromising on tumour control [105]. The mechanism behind the FLASH is not fully understood yet, although it is thought that the oxygen depletion hypothesis can help to explain it. Here, FLASH induces local oxygen depletion which generates a transient state of radiation-induced hypoxia and radioresistance within the tumour and surrounding normal tissue. This initial assumption appears counterintuitive for tumour control given how hypoxia induces radioresistance, however this effect on the tumour is predicted to be minimal whereas comparatively this will induce a high degree of normal tissue sparing [106]. Although the complete mechanism is still to be elucidated, differences in oxidative metabolism in tumour cells compared to normal cells is also suggested to play a role [107]. More substantial research is required to validate this as the major mechanism, and particularly in appropriate HNSCC tumour models to fully understand the radiobiology and future clinical application of FLASH radiotherapy.

## 6. Conclusions and Outlook

The negative impact that hypoxia has on radiotherapy effectiveness is widely accepted, although recently there has been a significant lack in the advancement of strategies to overcome this phenotype. Mechanistically, there is a solid understanding of the oxygen fixation hypothesis, but on a molecular level, there remain gaps to be filled. Recent work revealing the role of HIF in the cellular response to hypoxia has advanced our understanding and provided a greater insight into the complex signalling pathways and cellular responses that occur in hypoxic environments. However, the impact that these hypoxic responses have on radioresistance in tumour models such as HNSCC remains unclear. As a consequence, cellular proteins and pathways that could be targeted to overcome hypoxia-induced radioresistance are uncertain, although strategies targeting HIF or utilising immunotherapy and high-LET radiation could be explored further. Specifically, it is clear that preclinical research on hypoxia in HNSCC would benefit from the use of 3D models, such as spheroids and patient-derived organoids, that offer a more accurate tumour model as they contain oxygen gradients, which are more clinically relevant to the hypoxic nature of patient tumours. These also allow the oxygen concentration to be more dynamic and differ from the static single oxygen tensions commonly used in vitro using immortalised cell lines. Furthermore, a more detailed examination of acute and long-term hypoxia needs to be performed as this can have profound effects on adaptive cellular responses and especially in terms of acquiring radioresistance.

Translating hypoxic radiosensitisation strategies from in vitro preclinical research into clinical trials is slow, and with high study failures. A key reason for this clinical translation failure could be attributed to a lack of distinction between patients who would benefit from hypoxic modification, and those who would not. The inclusion of patients with mild hypoxic tumours and subjecting them to hypoxic modification prior to radiotherapy would likely negatively impact the success of the study as these individuals would not benefit from such any approach. Future clinical trials should therefore focus on differentiating between patients with severely hypoxic tumours and those with milder hypoxic conditions. This is undoubtedly challenging given the limited non-invasive techniques available to monitor tumour hypoxia, but which would nevertheless provide more accurate conclusions on the success of hypoxic modification techniques relative to the degree of hypoxia. The scope of the issue is not underestimated, but it is vital that we continue both preclinical and clinical research to understand the molecular and cellular contributions towards hypoxic radioresistance in HNSCC. Improving the effectiveness of radiotherapy in HNSCC patients will not only contribute to better patient prognosis, but also improve quality of life.

## Figures and Tables

**Figure 1 cancers-14-04130-f001:**
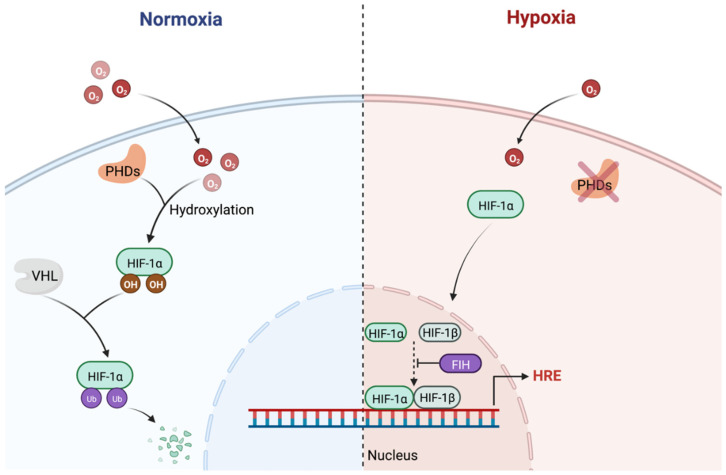
The HIF degradation pathway. Under normoxic conditions, HIF-1α undergoes hydroxylation via PHDs which require oxygen. This hydroxylation allows HIF to be preferentially recognised by the VHL protein which targets the protein for ubiquitylation-dependent proteasomal degradation. In hypoxic environments, HIF-1α is not targeted for hydroxylation and degradation, therefore accumulates and is translocated into the nucleus. Once in the nucleus, it can form a heterodimer with HIF-1β which binds to the hypoxia response elements (HRE) on the target genes to initiate their transcription. Figure created with BioRender.com (accessed on 13 July 2022).

**Figure 2 cancers-14-04130-f002:**
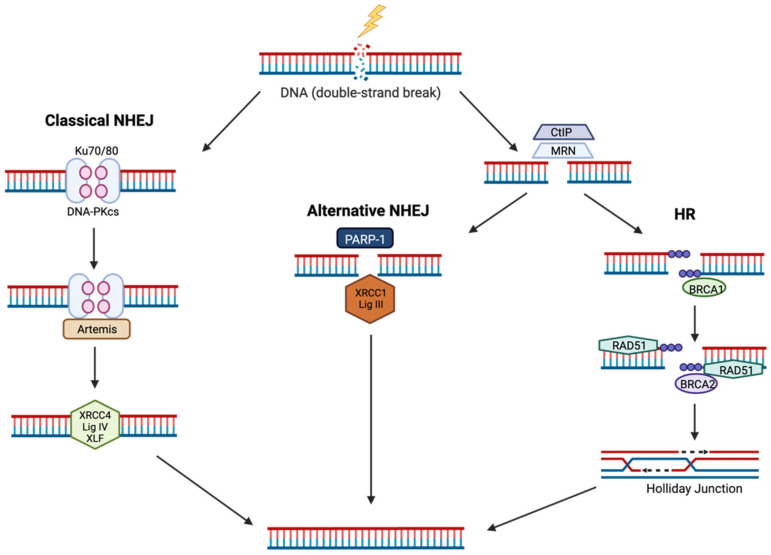
DSB repair mechanisms following IR. NHEJ can be divided into classical and alternative pathways, whereby classical NHEJ utilises Ku70/80 that binds to the DSB ends and recruits DNA-PKcs, Artemis and XRCC4/LIG IV to complete DNA ligation. Alternative NHEJ utilises the MRN-CtIP complex for DSB end resection following PARP-1 binding, and DNA ligation performed via XRCC1-LIG IIIα complex. During HR, DSB ends are resected via the MRN-CtIP complex and promotes BRCA1, RPA and RAD51 binding. This promotes strand invasion in a BRCA2 dependent manner, followed by the formation and resolving of Holliday junctions. Figure created with BioRender.com.

**Table 1 cancers-14-04130-t001:** Clinical trials performed to overcome hypoxic radioresistance in HNSCC.

Modification	Response	References
Hyperbaric oxygen	Many studies have utilised hyperbaric oxygen administer to HNSCC patients prior to radiotherapy. Safety implications have hindered its progression.	[65,66,67,68,69,70,71]
Carbogen	Carbogen breathing (95% oxygen and 5% carbon dioxide) prior to radiotherapy did not improve local or regional control of HNSCC patient tumours.	[72]
ARCON	Accelerated radiotherapy with carbogen and nicotinamide (ARCON) trials in HNSCC patients generally improved regional tumour control but no benefit on local control.	[73,74]
Nitroimidazoles	Misonidazole and Etanidazole—no differences in tumour control when combined with radiotherapy. Severe side effects were also reported. Nimorazole—increased tumour control when combined with radiotherapy in HNSCC patients but only those with confirmed tumour hypoxia benefitted.	[75,76,77,78,79,80,81]
Tirapazamine	Phase II clinical trials showed promising results for HNSCC patients. However, the success was not sustained into phase III.	[82,83]

**Table 2 cancers-14-04130-t002:** Current and future strategies to overcome hypoxia induced radioresistance in HNSCC.

Target Strategy	Comments	References
DDR	Alterations in the DDR in hypoxic tumour cells have been observed, so DDR could be a suitable target. For example, ATR, DNA-PKcs and PARP inhibitors should be explored more in hypoxic HNSCC models.	[84,85,86,87,88,89,90]
HIF	HIF overexpression linked to poor prognosis of HNSCC, so remains an attractive target. Limited reported evidence on the impact of HIF inhibition on HNSCC radioresistance.	[91,92,93,94,95]
Immunotherapy	Association between PDL-1 and HIF-1α, and that hypoxia may alter tumour immunosurveillance. Possibility for targeting both HIF-1α and PDL-1 to overcome hypoxic radioresistance. Clinical trials in HNSCC combining immunotherapy with radiotherapy are ongoing.	[96,97,98,99,100]
High LET and FLASH radiotherapy	High-LET radiotherapy reduces the need for oxygen within the tumour for effectiveness, so has the potential to overcome hypoxic radioresistance. However, evidence of high-LET and FLASH radiotherapy in HNSCC is lacking.	[101,102,103,104,105,106,107]

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
