# Peer review of "Overcoming the Impact of Hypoxia in Driving Radiotherapy Resistance in Head and Neck Squamous Cell Carcinoma"

_cancers, 2022, doi:10.3390/cancers14174130_

Round 1

Reviewer 1 Report

This is a timely, accurate, and comprehensive review on the role of hypoxia in radiotherapy response in HNSCC, including exploring therapeutic approaches to induce radio sensitisation in hypoxia. I would recommend this review for publication, with some minor comments (noted below) to be addressed.

In section 5.4, it is important to define more clearly the impact of low oxygen tensions in DDR activation etc, and the term 'radiobiological hypoxia' should be used to note the severe hypoxic conditions where clearer radiotherapy resistance is observed. In this same section, it would be important to note studies where DDR inhibitors have been used as hypoxia radiosensitisers in radiobiological hypoxic conditions, albeit in other tumours types, as proof of principle, especially studies where ATR inhibitors and PARP inhibitors have been used.

In section 5.7, the authors should expand on the currently hypothesised mechanisms regarding how the FLASH effect leads to different impact in tumour vs normal tissues leading to normal tissue sparing, linked with the noted oxygen depletion hypothesis highlighted in the text. It is appreciated this process is complex and still not fully understood, but some clarification would improve this section.

Reviewer 2 Report

Hypoxia is very common in most solid tumors and is a driving force for malignant progression as well as radiotherapy and chemotherapy resistance. In this manuscript, the authors provided an overview of how hypoxia alters molecular and cellular processes contributing to radioresistance, particularly in HNSCC, and what strategies have and could be explored to overcome hypoxia-induced radioresistance. Their work could be of great value for a better understanding of the development of novel drugs for radiotherapy treatment. However, some issues should be addressed to strengthen this manuscript. 

One more table is needed to summarize the strategies for overcoming hypoxia-induced radioresistance.

The title of this manuscript is “The Impact of Hypoxia in Driving Radiotherapy Resistance”. In the second paragraph of "3.1 Hypoxia and radiotherapy", the discussion of the correlation between hypoxia and radioresistance is quite important to fit this title. What kind of DNA damage and its effect? More discussion is needed. Specifically, I suggest that the authors add and update the reference. e.g., two important ROS-induced DNA damages are missed: Chemical science, 2019, 10(15): 4272-4281; J. Am. Chem. Soc. 2022, 144, 1, 454–462. In addition, the definition and corresponding outcome of oxygen fixation are unclear here.

This manuscript has abundant content that discusses the strategies for overcoming hypoxia-induced radioresistance, it should be present in the title to attract more readers.

Reviewer 3 Report

In this review, Hill et al discuss our current understanding of how hypoxia leads to radioresistance in head and neck squamous cell carcinoma (HNSCC) and discuss the strategies to overcome this. The review nicely summarizes a number of aspects of the general topic and will point readers to more in-depth reviews and primary literature. As such, it is a valuable contribution for a general audience.

A few comments are listed below.

1.      In the section “Overcoming hypoxic radioresistance in HNSCC”, the authors talk about targeting the DDR by PARPi to overcome chemoresistance. However, a recent study showed that moderate hypoxia promotes PARPi resistance in both HR-proficient and -deficient cancer cells by reducing ROS-induced DNA damage. Addition of these findings to the review would be beneficial. 

2.      The authors discuss the cellular DNA damage response to IR and talk about repair pathways including NHEJ and HR. Interestingly, hypoxia was recently shown to cause the loss of HR proteins and activation of the ROS-ATM-MRE11 cascade (Somyajit et al, 2021). The authors could consider including these findings in the review.  
